# AUTOENCODERS FOR ANOMALY DETECTION ARE UNRELIABLE

## ABSTRACT

Autoencoders are frequently used for anomaly detection, both in the unsupervised and semi-supervised settings. They rely on the assumption that when trained using the reconstruction loss, they will be able to reconstruct normal data more accurately than anomalous data. Some recent works have posited that this assumption may not always hold, but little has been done to study the validity of the assumption in theory. In this work we show that this assumption indeed does not hold, and illustrate that anomalies, lying far away from normal data, can be perfectly reconstructed in practice. We extend the understanding of autoencoders for anomaly detection by showing how they can perfectly reconstruct out of bounds, or extrapolate undesirably, and note how this can be dangerous in safety critical applications. We connect theory to practice by showing that the proven behavior in linear autoencoders also occurs when applying non-linear autoencoders on both tabular data and real-world image data, the two primary application areas of autoencoders for anomaly detection.

## 1 INTRODUCTION

Autoencoders are one of the most popular architectures within anomaly detection, either directly, or as a scaffold or integral part in larger pipelines or architectures. They are commonly used across a variety of domains, such as predictive maintenance (Kamat & Sugandhi, 2020), network anomaly detction (Said Elsayed et al., 2020), and intrusion detection (Farahnakian & Heikkonen, 2018), but find much contemporary use in computer vision anomaly detection, with applications such as industrial inspection (Tsai & Jen, 2021), medical imaging (Wei et al., 2018), structural health monitoring (Chow et al., 2020) and video surveillance (Zhao et al., 2017; Cruz-Esquivel & Guzman-Zavaleta, 2022). Many of these applications are safety critical, meaning that the reliability of these algorithms is of utmost importance in order to prevent catastrophic failure and associated dangers and consequences.

Anomaly detection using autoencoders typically relies on using the reconstruction loss, often the mean squared error (MSE), as a proxy for "anomalousness". The underlying assumption is that anomalies are harder to reconstruct, and will therefore have a higher reconstruction loss. However, the validity of this assumption has been questioned in recent years. Merrill & Eskandarian (2020) and Beggel et al. (2020) for example state that anomalies in the training data might lead to reconstruction of anomalies, leading to unreliable detectors. Some researchers have noted that reconstruction of unseen anomalies might also occur in the semi-supervised setting, where the training data is assumed to contain no anomalies (Astrid et al., 2021; 2024; Gong et al., 2019; Zong et al., 2018). Yet, little work has been done on the nature of failure and reliability of autoencoders beyond experimental evaluation, leaving a gap in theoretical rigor.

In this work we provide valuable insights into the reliability of autoencoders for anomaly detection. Following the seminal works of Bourlard & Kamp (1988) and Baldi & Hornik (1989) we develop a theory on autoencoder failure modes, whilst briefly examining how various activation functions influence these failures. We show that this theory is not just a rarely occurring edge case, but also show failure cases on tabular data and on real-world image data commonly used in anomaly detection benchmarks. By doing this we provide a foundation for future research in solving the demonstrated unreliability of autoencoders for anomaly detection and furthering our understanding of autoencoders. We how that for different architectures and activations functions, even with

sufficiently constrained latent spaces, these problems persist. To ensure reproducibility of all experiments we use only open-source data and provide code for all experiments[1].

## 2 RELATED WORK

This work is not the first to recognize that autoencoders have several issues as anomaly detectors. The most discussed type of failure is the unwanted reconstruction of anomalies, which is also the focus of this work. Several causes of this unwanted behavior have been proposed.

Many works focus on the unsupervised setting, and observe that contrary to prior belief, autoencoders will fairly easily reconstruct any anomalies present in the training data, leading to an unusable detector (Merrill & Eskandarian, 2020; Beggel et al., 2020; Cheng et al., 2021; Tong et al., 2022).

Several works cover the anomaly reconstruction problem within the semi-supervised setting. Most commonly, it is only experimentally observed that anomalies are well reconstructed (Gong et al., 2019; Zong et al., 2018; Cheng et al., 2021; Astrid et al., 2021; 2024; Salehi et al., 2021; Nalisnick et al., 2019; Lyudchik, 2016). Based on these experimental results, some solutions have been proposed. Gong et al. (2019) mention that out-of-bounds reconstruction can occur and propose adding a memory module to the autoencoder to alleviate the issue. While the addition of the memory module can aid in limiting out-of-bounds reconstruction, it also leads to a severely decreased reconstruction ability and substantial added complexity in training and optimizing the network. Zong et al. (2018) note that while some anomalies have a high reconstruction loss, some occupy the region of normal reconstruction loss, and add a Gaussian mixture model to the latent space to aid in detection of anomalies under the assumption that anomalies occupy a low-density region in the latent space. Similarly, Cheng et al. (2021) aim to detect anomalies in the latent space by looking at the distance to the centroid. From our experiments we can glean that relying on distances or densities in the latent space does not always work in practice. Astrid et al. (2021; 2024) make use of constructed pseudo anomalies in training the autoencoder. They add adaptive noise to normal samples to generate pseudo anomalies. In the reconstruction, they then optimize the reconstruction loss between the pseudo anomaly and the normal sample used to generate it. While they show promising results and greater discriminative power on benchmark datasets, they do not quantify to which degree their performance gains can be attributed to the reduction of the out-of-bounds reconstruction. Salehi et al. (2021) aim to limit the reconstruction of anomalies by generating adversarial samples. The adversarial examples are generated by perturbing the normal samples, and minimizing the effect those perturbations have in the latent space. This is similar to the concept of denoising autoencoders. Based on our experiments, we do not think this results in a reliable autoencoder, as often adversarial anomalies can occupy the latent space close to normal data.

Some authors have moved beyond the experimental, and propose explanations for the anomaly reconstruction problem. For example You et al. (2022), Lu et al. (2023) and Bercea et al. (2023) propose that anomaly reconstruction can happen because an autoencoder can learn an "identical shortcut", where both normal data and anomalous data is effectively reconstructed using an identity mapping. This point has however been countered by Cai et al. (2024) who show that by constraining the latent space to be sufficiently low dimensional, this problem can be avoided.

The second line of thought follows from VAE anomaly detection, where Havtorn et al. (2021) theorize that in out-of-distribution detection, unwanted reconstruction can happen due to a high correlation between learned low-level features for in- and out-of-distribution data.

A third line of thought is proposed by Zhou (2022) who propose that reconstruction of out-of-distribution samples can happen due to out-of-distribution data having smaller neural activations than in-distribution data.

Finally, some works theorize that autoencoders can perfectly reconstruct data due to the anomalies occupying the reconstruction hyperplane, or latent space manifold. Denouden et al. (2018) for example note this phenomenon, and aim to solve it by adding the Mahalanobis distance in the latent space to the loss. The most detailed work is that of Yoon et al. (2021) who provide an example of the hyperplane interpolating between clusters of data. They solve this by introducing a normalized

---

[1]Link to GitHub page: (Removed during review, copy of anonymized repo has been added as supplementary material.

autoencoder which reinterprets the autoencoder as a likelihood-based energy model. We specifically follow up on this line of reasoning, and provide mathematical proofs and experimental evidence of anomaly reconstruction due to both unwanted extrapolation and inter-class interpolation.

# 3 BACKGROUND

## 3.1 ANOMALY DETECTION

In practical anomaly detection, we attribute a score $s_i = f_{\text{anomaly score}}(\boldsymbol{x}_i)$ for each sample $\boldsymbol{x}_i \in \mathcal{X} = \mathbb{R}^n$, i.e. the $i$-th row of dataset $\boldsymbol{X}$ with size $m$-by-$n$. The score should then be higher for anomalous samples than for normal data. When applied to some dataset consisting of both normal data and anomalies, i.e. $\boldsymbol{X} = \{\boldsymbol{X}^{\text{normal}}, \boldsymbol{X}^{\text{anomalous}}\}$, a perfect anomaly detector will assign higher scores to anomalies than to normal data: $\min_i(f_{\text{anomaly score}}(\boldsymbol{x}_i^{\text{anomalous}})) > \max_i(f_{\text{anomaly score}}(\boldsymbol{x}_i^{\text{normal}}))$.

The two most common anomaly detection settings are unsupervised and semi-supervised. Unsupervised anomaly detection is characterized by having no discernible "train" and "test" splits. Instead, we only consider a single dataset $\boldsymbol{X} = \{\boldsymbol{X}^{\text{normal}}, \boldsymbol{X}^{\text{anomalous}}\}$, where we are uncertain which samples are anomalous and which are not. In the semi-supervised setting we instead have a conventional "train" and "test" set. The train set consists out of only normal samples: $\boldsymbol{X}^{\text{train}} = \{\boldsymbol{X}^{\text{train, normal}}\}$, while the test set is unknown, and can contain both normal and anomalous samples: $\boldsymbol{X}^{\text{test}} = \{\boldsymbol{X}^{\text{test, normal}}, \boldsymbol{X}^{\text{test, anomalous}}\}$. In this paper we will only consider the semi-supervised case, and simplify the notation with $\boldsymbol{X} = \boldsymbol{X}^{\text{train}}$ and $\boldsymbol{x}_i$ referring to an individual training data point, which in the semi-supervised case by definition is not an anomaly. We then consider a new data point $\boldsymbol{a}$ to determine whether this is an anomaly or not. In older literature, semi-supervised anomaly detection is often called one-class classification.

# 4 OUT-OF-BOUNDS RECONSTRUCTION

In this section we will show that autoencoders can yield zero-loss reconstruction far away from all training data, and that these autoencoders will then fail to detect certain anomalies. We will build our theory following the results of Bourlard & Kamp (1988), moving from PCA to linear autoencoders to non-linear autoencoders.

Out-of-bounds reconstruction is unwanted within the application of anomaly detection, as it leads to a low reconstruction loss for data that can be considered anomalous, thereby leading to false negatives. These regions of out-of-bounds reconstruction can also be exploited by targeted adversarial evasion attacks.

In the worst case, unwanted perfect reconstruction causes an anomaly $\boldsymbol{a} \in \mathbb{R}^n$ far from all training data to be ranked as being less anomalous than or equally anomalous as all training data, that is: $f_{\text{anomaly score}}(\boldsymbol{a}) \leq \min_i(f_{\text{anomaly score}}(\boldsymbol{x}_i))$.

## 4.1 ANOMALY DETECTION USING THE RECONSTRUCTION LOSS

Both PCA and autoencoders are dimensionality reduction techniques that can be used to detect anomalies using their reconstruction loss, commonly known as the mean squared error, or MSE. We can calculate the reconstruction loss by comparing a sample $\boldsymbol{x}_i$ to its reconstruction $\hat{\boldsymbol{x}}_i$: $\mathcal{L}_R(\boldsymbol{x}_i, \hat{\boldsymbol{x}}_i) = \frac{1}{n}\sum_{j=1}^{n}(x_{i,j} - \hat{x}_{i,j})^2$, for each sample vector $\boldsymbol{x}_i$. This reconstruction loss often serves as a proxy for detecting anomalies, with the underlying assumption that a higher reconstruction loss indicates a higher likelihood of the sample being an anomaly.

For both PCA and autoencoders we want to find a lower-dimensional encoding $\boldsymbol{Y}$, e.g. $d < n$, in the encoding space $\mathcal{Y} = \mathbb{R}^d$ by applying the function $g : \mathcal{X} \to \mathcal{Y}$. We then decode $\boldsymbol{Y}$ by transforming it back into the space $\mathcal{X}$ through the decoder $h : \mathcal{Y} \to \mathcal{X}$, yielding the reconstructed data $\hat{\boldsymbol{X}}$. Summarizing, we learn the concrete transformations $\boldsymbol{X} \xrightarrow{g} \boldsymbol{Y} \xrightarrow{h} \hat{\boldsymbol{X}}$.

We can then formulate the anomaly scoring function in terms of the reconstruction loss, encoder, and decoder: $f_{\text{anomaly score}}(\boldsymbol{x}_i) = \mathcal{L}_R(\boldsymbol{x}_i, h(g(\boldsymbol{x}_i)))$. The worst case can then be formulated as: there exists an $\boldsymbol{a}$ far from all training data such that $\mathcal{L}_R(\boldsymbol{a}, \hat{\boldsymbol{a}}) \leq \min_i(\mathcal{L}_R(\boldsymbol{x}_i, \hat{\boldsymbol{x}}_i))$.

## 4.2 PCA

In PCA, we factorize $X$ as $X = U\Sigma V^T$ using singular value decomposition (SVD), where $U$ and $V$ are orthonormal matrices containing the left- and right-singular vectors of $X$, respectively, and $\Sigma$ is a diagonal scaling matrix. The encoding, or latent, space is then obtained by projecting onto the first $d$ right-singular vectors: $Y = g(X) = XV_d$, where $V_d$ contains the first $d$ columns of $V$. The transformation back into $\mathcal{X}$ is given by $\hat{X} = h(Y) = YV_d^T$.

We will show that there exist some $a \in \mathcal{R}^n$ for which the reconstruction loss is zero, but that are far away from the normal data, i.e. $\min_i(\text{dist}(x_i, a)) > \delta$, for any arbitrary choice of $\delta$. Hereby we prove that it is possible to find anomalous, adversarial, examples with perfect out-of-bounds reconstruction. We can prove this even in the semi-supervised setting, where we guarantee that the model was not exposed to anomalous data at training time. Due to the semi-supervised setting being more restrictive, the proofs also apply to the unsupervised case.

**Lemma 1.** *Let* $a \in \mathbb{R}^n$. *If* $a$ *lies in the column space of* $V_d$, *then the reconstruction loss* $\mathcal{L}_R(a, h(g(a))) = 0$.

*Proof.* To prove this, we need to show that there exists some $a$ such that $h(g(a)) = a$. For PCA, this condition can be written as:
$$aV_dV_d^T = a.$$

Assume $a$ is in the row space of $V_d^T$. Then $a$ can be expressed as a linear combination of the rows in $V_d^T$. Let $c \in \mathbb{R}^d$ be such that:
$$a = cV_d^T.$$

Substitute $a$ into the left-hand side of the reconstruction equation:
$$aV_dV_d^T = cV_d^TV_dV_d^T.$$

Since $V_d$ is composed of orthonormal columns, $V_d^TV_d = I_d$, where $I_d$ is the $d$-by-$d$ identity matrix. Therefore:
$$cV_d^TV_dV_d^T = cV_d^T = a.$$

Thus, $a$ satisfies the condition $h(g(a)) = a$, implying that the reconstruction loss $\mathcal{L}_R(a, g(h(a))) = 0$. $\square$

**Theorem 2.** *There exists some adversarial example* $a \in \mathbb{R}^n$ *that is far from all normal data, i.e.* $\min_i(\text{dist}(x_i, a)) > \delta$, *for an arbitrary* $\delta$ *and the Euclidean distance metric, but still has a reconstruction loss* $\mathcal{L}_R(a, g(h(a))) = 0$.

*Proof.* The lemma demonstrates that any $a$ in the column space of $V_d$ will have zero reconstruction loss.

If we then define $a = x_iV_dV_d^T + cV_d^T$, $a$ will still have zero reconstruction loss.

Then for the Euclidean distance it follows that:
$$\text{dist}(x_i, a)^2 = \|x_i - x_iV_dV_d^T\|^2 + \|a - x_iV_dV_d^T\|^2,$$

or the squared Euclidean distance is equal to the distance from $x_i$ to its projection onto the hyperplane $x_iV_dV_d^T$ plus the distance from that projection $x_iV_dV_d^T$ to $a$.

It then follows that:
$$\text{dist}(x_i, a)^2 \geq \|a - x_iV_dV_d^T\|^2 = \|x_iV_dV_d^T + cV_d^T - x_iV_dV_d^T\|^2 = \|cV_d^T\|^2,$$

which we can increase to arbitrary length. This can be intuited as moving the projection of $x_i$ along the hyperplane.

To ensure that we increase the distance to all points $x_i$ rather than just a single one, we need to move outward starting from a sample on the convex hull enclosing $XV_d$. Any point in this convex set, that is the set of points occupying the convex hull, can be moved along the hyperplane to increase the distance to all points $x_iV_d$, and therefore to all points $x_i$ as long as $c$ lies in the direction from $x_iV_d$ to the exterior of the convex hull.

We can thus always find a $\boldsymbol{a} = \boldsymbol{x}_i \boldsymbol{V}_d \boldsymbol{V}_d^T + \boldsymbol{c} \boldsymbol{V}_d^T$, for some $\boldsymbol{x}_i \boldsymbol{V}_d$ in the convex set of $\boldsymbol{X} \boldsymbol{V}_d$ and choose $\boldsymbol{c}$ so that it points from $\boldsymbol{x}_i \boldsymbol{V}_d$ to the exterior of the convex hull and is of sufficient length such that $\min_i(\text{dist}(\boldsymbol{x}_i, \boldsymbol{a})) > \delta$, for an arbitrary $\delta$.

Hence, all vectors $\boldsymbol{a} \in \mathbb{R}^n$ found in this way are constructed anomalies, or adversarial examples, that are far from all normal data, but still have zero reconstruction loss. □

We posit that the same principle applies to other distance metrics, and the intuition is that this follows the same line of reasoning as presented here for the Euclidean distance.

## 4.3 Linear Autoencoders

Linear neural networks, like PCA, can also exhibit out-of-bounds reconstruction for certain anomalous data points.

Linear autoencoders consist of a single linear encoding layer and a single linear decoding layer. Given a mean-centered dataset $\boldsymbol{X}$, the encoding and decoding transformations can be represented as follows:

$$\boldsymbol{Y} = g(\boldsymbol{X}) = \boldsymbol{X} \boldsymbol{W}_{\text{enc}},$$

$$\hat{\boldsymbol{X}} = h(\boldsymbol{Y}) = \boldsymbol{Y} \boldsymbol{W}_{\text{dec}}^T = \boldsymbol{X} \boldsymbol{W}_{\text{enc}} \boldsymbol{W}_{\text{dec}}^T$$

where $\boldsymbol{W}_{\text{enc}}$ is the $n$-by-$d$ weight matrix of the encoder, and $\boldsymbol{W}_{\text{dec}}^T$ is the $d$-by-$n$ weight matrix of the decoder. We assume the autoencoder to have converged to the global optimum. Note that we define $\boldsymbol{W}_{\text{dec}}^T$ in its transposed form to illustrate its relation to $\boldsymbol{V}_d^T$. Due to the full linearity of the model, even multiple layer networks can be simplified to a single multiplication. It is known that a well-converged linear autoencoder finds an encoding in the space spanned by the first $d$ principal components (Baldi & Hornik, 1989). In other words, the encoding weight matrix can be expressed in terms of the first $d$ principal components and some invertible matrix $\boldsymbol{C}$:

$$\boldsymbol{W}_{\text{enc}} = \boldsymbol{V}_d \boldsymbol{C}.$$

At the global optimum $\boldsymbol{W}_{\text{dec}}^T$ can be expressed as the inverse of $\boldsymbol{W}_{\text{enc}}$ :

$$\boldsymbol{W}_{\text{dec}}^T = \boldsymbol{W}_{\text{enc}}^{-1} = (\boldsymbol{V}_d \boldsymbol{C})^{-1} = \boldsymbol{C}^{-1} \boldsymbol{V}_d^{-1} = \boldsymbol{C}^{-1} \boldsymbol{V}_d^T.$$

To show that linear autoencoders can exhibit perfect out-of-bounds reconstruction, we then prove the same lemma as for PCA.

**Lemma 3.** *Let $\boldsymbol{a} \in \mathbb{R}^n$. If $\boldsymbol{a}$ lies in the column space of $\boldsymbol{V}_d$, then the reconstruction loss $\mathcal{L}_R(\boldsymbol{a}, h(g(\boldsymbol{a}))) = 0$.*

*Proof.* We need to show that there exists some $\boldsymbol{a}$ such that $h(g(\boldsymbol{a})) = \boldsymbol{a}$. For linear autoencoders, this condition can be written as:

$$\boldsymbol{a} \boldsymbol{W}_{\text{enc}} \boldsymbol{W}_{\text{dec}}^T = \boldsymbol{a}.$$

Assume $\boldsymbol{a}$ is in the row space of $\boldsymbol{V}_d^T$. Then $\boldsymbol{a}$ can be expressed as a linear combination of the rows in $\boldsymbol{V}_d$. Let $\boldsymbol{c} \in \mathbb{R}^d$ be such that:

$$\boldsymbol{a} = \boldsymbol{c} \boldsymbol{V}_d^T.$$

Then it follows that:

$$\boldsymbol{a} \boldsymbol{W}_{\text{enc}} \boldsymbol{W}_{\text{dec}}^T = \boldsymbol{c} \boldsymbol{V}_d^T \boldsymbol{W}_{\text{enc}} \boldsymbol{W}_{\text{dec}}^T = \boldsymbol{c} \boldsymbol{V}_d^T \boldsymbol{V}_d \boldsymbol{C} \boldsymbol{C}^{-1} \boldsymbol{V}_d^T = \boldsymbol{c} \boldsymbol{V}_d^T \boldsymbol{V}_d \boldsymbol{V}_d^T = \boldsymbol{c} \boldsymbol{V}_d^T = \boldsymbol{a},$$

indicating that $\boldsymbol{a}$ satisfies the condition $h(g(\boldsymbol{a})) = \boldsymbol{a}$, implying that the reconstruction loss $\mathcal{L}_R(\boldsymbol{a}, g(h(\boldsymbol{a}))) = 0$. □

After proving this lemma, the case of the linear autoencoder reduces to that of PCA, with the same proof that adversarial examples satisfying $\min_i(\text{dist}(\boldsymbol{x}_i, \boldsymbol{a})) > \delta$, with a reconstruction loss $\mathcal{L}_R(\boldsymbol{a}, g(h(\boldsymbol{a}))) = 0$, exist.

An extension of this proof to the case of linear networks with bias terms applied on non-centered data can be found in Appendix A.1.

## 4.4 Non-Linear Autoencoders

In this section we show that datasets exist for which we can prove that non-linear neural networks perform the same unwanted out-of-bounds reconstruction. Then we experimentally show that this behavior indeed occurs in more complex real-world examples.

### 4.4.1 Failure of a Non-Linear Network with ReLU Activations

We can show on a simple dataset that unwanted reconstruction behavior can occur in non-linear autoencoders. We consider a two-dimensional dataset $\boldsymbol{X}$ consisting out of normal samples $\boldsymbol{x}_i = \alpha_i(1, 1)$, where $\alpha_i$ is some scalar. Simply put, every normal sample $\boldsymbol{x}_i$ occupies the diagonal. This dataset can be perfectly reconstructed by a linear autoencoder with $\boldsymbol{W}_{\text{enc}} = \beta(1, 1)^T$, where $\beta$ is some scalar. The simplest non-linear autoencoder with ReLU activation will find the same weights, but with a bias such that $\boldsymbol{x}_i \boldsymbol{W}_{\text{enc}} + \boldsymbol{b}_{\text{enc}} > 0$ for all $\boldsymbol{x}_i \in \boldsymbol{X}$, i.e. $\boldsymbol{b}_{\text{enc}} \geq \min_i(\boldsymbol{x}_i \boldsymbol{W}_{\text{enc}})$. This will then lead to a perfect reconstruction for all $\boldsymbol{x}_i$. Adversarial anomalies $\boldsymbol{a}$ can also be easily found as $\boldsymbol{a} = c(1, 1)$, where $c \gg \frac{\max_i(\boldsymbol{x}_i)}{(1, 1)}$ is some sufficiently large scalar such that $\min_i(\text{dist}(\boldsymbol{x}_i, \boldsymbol{a})) > \delta$ is satisfied. We theorize that even beyond this simple case, similar linear behavior can occur beyond the convex hull that the normal data occupies. We experimentally show this anomaly reconstruction behavior in later sections.

### 4.4.2 Tabular Data

On more complex datasets, we observe similar behaviour. We have synthesized several two-dimensional datasets to show how non-linear autoencoders behave when used for anomaly detection. These datasets, as well as contours of the MSE of autoencoders trained on these datasets, are visualized in Figure 1. In each of these cases, we have synthesized a dataset by sampling 100 points per distribution, either from a single or from two Gaussians as in 1a, 1b, 1e, and 1f, or from $\boldsymbol{x}_2 = \boldsymbol{x}_1^2$ with added Gaussian noise in 1c and 1d. In all cases we use an autoencoder with layer sizes [2,5,1,5,2], except for 1d, where we use layer sizes of [2,100,20,1,20,100,2] to better model the non-linearity of the data. All layers have ReLU (Subfigures 1a, 1b, 1c, and 1d ), or sigmoid (Subfigures 1e and 1f) activations, except for the last layer, which has a linear activation. In these figures the color red is used to highlight those areas where the autoencoder is able to nearly perfectly reconstruct the data, i.e. MSE $< \epsilon = 0.1$.

**ReLU Activation Failure on Tabular Data** We can readily observe some of the problematic behavior of autoencoders as anomaly detectors. Firstly, in Figure 1a we observe that well outside the bounds of the training data there is an area with a near-perfect reconstruction. Worryingly, the reconstruction loss is lower than for a large part of the area which the normal data occupies. If we move in the $(-1, -1)$ direction, the encoder and decoder will no longer match perfectly. Even so, problematically low reconstruction losses can be found in this direction. In Figures 1c and 1d we see the same linear out-of-bounds behavior. In each of these cases, the mismatch between encoder and decoder in the linear domain is less noticeable, leading to even larger areas of near-perfect reconstruction. Lastly, in Figure 1b that there is an area between the two clusters with a good reconstruction. Likely the units responsible for this area are still close to their initialization, and due to the simple shallow network architecture can not meaningfully contribute to the reconstruction of any samples.

Our intuition of this behavior directly relates to the proof of out-of-bounds reconstruction we have provided for linear autoencoders. At the edge of the data space, only a few of the ReLU neurons activate. Beyond this edge, no new neurons will activate, nor will any of the activated ones deactivate. This can lead to linear behavior on some edge of the data space, i.e., in this area the network reduces to a linear transformation $\boldsymbol{W}_{\text{enc}}$. If we now synthesize some $\boldsymbol{a}$ such that it lies in the column space of $\boldsymbol{W}_{\text{enc}}$, we can again find some adversarial anomalies $\boldsymbol{a} = c\boldsymbol{W}_{\text{enc}}^T$. Like we have observed in Figure 1a, there may be a mismatch between the encoder and decoder, even at the global optimum, so we might not be able to increase $c$ towards infinity and still find adversarial examples with $\mathcal{L}_R(\boldsymbol{a}, g(h(\boldsymbol{a}))) < \epsilon$.

**Sigmoid Activation Autoencoders** Nowadays, full sigmoid networks have mostly fallen out of favor in deeper networks due to their vanishing gradients (Hochreiter, 1991; Glorot et al., 2011).

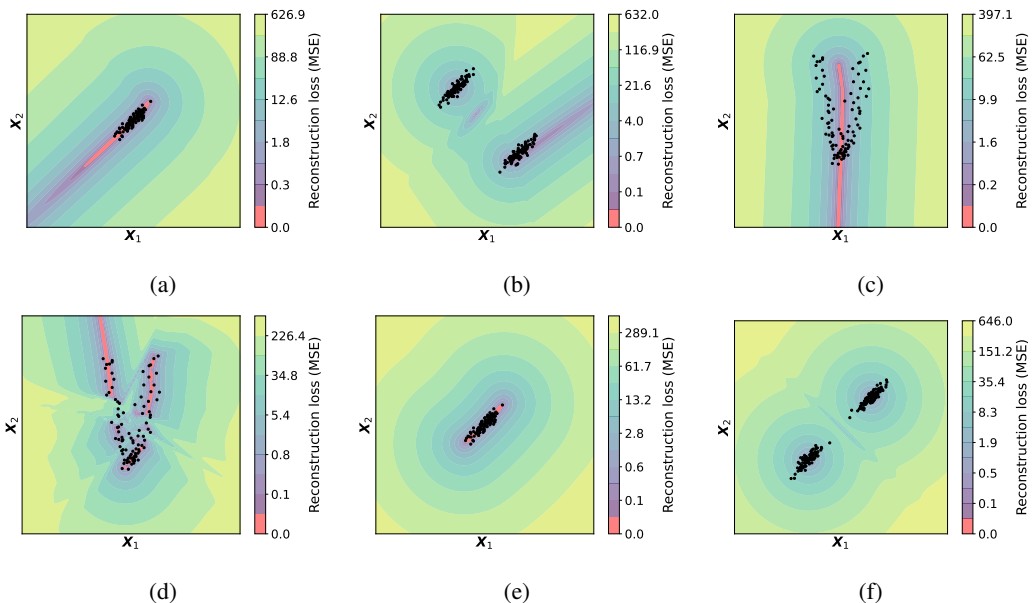

Figure 1: Plots of the contours of the reconstruction loss of non-linear autoencoders when applied to 3 distinct datasets. The datasets consist out of a 100 samples from a 2D Gaussian (a, e), 100 samples for each of 2 different 2D Gaussians (b, f), and a 100 samples from a classic banana shaped function with Gaussian noise (c, d). In (a,b,c,e,f) a [2,5,1,5,2] autoencoder is used, while for (d) a deeper [2,100,20,1,20,100,2] autoencoder is used. The contourplot is colored red whenever the MSE is below a set threshold $\epsilon < 0.1$ to indicate a near-perfect reconstruction. Note that the color scaling is exponential to better visualize the MSE loss.

However, sigmoids are more attractive to use in anomaly detection because they lead to networks that do not exhibit the hyperplane issues that the ReLU suffers from. While sigmoids have the more desirable property of tapering off at high or low input, making it hard to perfectly reconstruct data far away from normal data, autoencoders with just sigmoid activation can still behave unexpectedly, albeit less so than those with ReLU activation.

We can see in Figure 1e that the data is nicely approximated by a sigmoid autoencoder. It extends nicely to the first and last samples on the direction of the first principal component, and does not extend beyond that. When we extend this example to multimodal data, as in Figure 1f, we can see different undesirable behavior arising. There exists an area where the sigmoids reconstructing both clusters intersect. Due to the two distinct sigmoids mixing, we can find a hyperplane orthogonal to the first principal component where the reconstruction loss is much lower than would be expected. While in this case there are no points on the hyperplane which would have a lower reconstruction loss than all normal data, there is still a substantial area where the reconstruction loss is lower than for many of the normal data points.

**Other Activation Functions**   While we have explicitly discussed the ReLU and sigmoid activation functions, the behavior shown is similar for other activation functions. Effectively, we can categorize most activation functions as those having an order of continuity of $C^0$ like the ReLU, or $C^\infty$ like the sigmoid. In summary, activation functions with an order of continuity of $C^0$ suffer most from out-of-bounds reconstruction, but allow for more easily trainable deep networks. In contrast, activation functions with an order of continuity of $C^\infty$ generally have more desirable properties for anomaly detection, but are harder to use in deep networks due to the vanishing gradient.

### 4.5 CONVOLUTIONAL AUTOENCODERS

All the previous examples clearly illustrate autoencoders' possible failure and unreliability when used for anomaly detection on tabular data. Yet, many applications of anomaly detection are in

computer vision, where anomaly detection can be used to detect foreign objects. Typical examples of computer vision anomaly detection are surveillance, where videos are analyzed to find possible security threats (Nayak et al., 2021; Sultani et al., 2018), structural health monitoring (Bao et al., 2019), and industrial inspection (Bergmann et al., 2021).

For most applications of autoencoders on image data, the architecture is fairly straightforward. ReLU activation functions are most commonly used throughout the network, with a sigmoid activation at the final layer. Connections to and from the bottleneck layer are often chosen to be just linear, to allow for a richer internal representation. As most layers have ReLU activation functions, these networks do not suffer from the vanishing gradient. Yet, due to using a sigmoid at the last layer, these networks suffer less from the issues encountered in full ReLU/linear networks as discussed in Section 4.4.2. Nonetheless, we will show that even on more complex real-world problems, autoencoders remain unreliable and are often able to reconstruct out-of-bounds.

### 4.5.1 FAILURE ON REAL-WORLD DATA: MNIST

To show that deeper non-linear networks trained on real-world image data can still undesirably reconstruct anomalies we will study an autoencoder for anomaly detection that was trained on the well-known MNIST dataset (LeCun, 1998). Benchmarking computer vision anomaly detection algorithms is not as standardized as classification benchmarking, as datasets with "true" anomalies are exceedingly rare. The common method for benchmarking these algorithms is to take a classification dataset and select a subset of classes as "normal" data and another distinct subset as "anomalies". This is analogous to other, more-developed, fields such as tabular anomaly detection (Bouman et al., 2024). There is no general consensus on which digits are taken as the normal data, and how many. In our experiments, both shown and non-shown, we have tried several different combinations and observe that in some cases out-of-bounds reconstruction occurs.

The 2D convolutional autoencoder we will discuss has a 2-layer encoder and 2-layer decoder. Down- and upsampling to and from the latent space is done using a fully connected linear layer. The convolutional layers all use ReLU activations, except for the last one, which is a sigmoid to bound the data to the original 0-1 range. In these experiments, the latent space is set to be two dimensional, far below the maximum to avoid the "identical shortcut" as noted by Cai et al. (2024). This serves as proof that the "identical shortcut" is not the cause of anomaly reconstruction. In Figures 2a and 2b we show how the reconstruction loss behaves in the latent space when we apply this autoencoder on a train set consisting out of a subset of digits. These contourplots are constructed by sampling each point in the latent space, decoding it to get an artificial sample, and then calculating the reconstruction loss between the artificial sample and its reconstruction loss. We subsequently show the latent representations of all normal data in the same space. We should note that as the encoder is a many-to-one mapping, the reconstruction loss in the grid does not necessarily correspond to the reconstruction loss of a real sample occupying the same point in that grid.

Looking at Figure 2a we see that a 2D latent space is able to separate the digits 4 and 5, with 7 occupying the middle between the two classes. As expected, the reconstruction loss grows the larger the distribution shift becomes. However, the reconstruction loss landscape is fairly skewed, with the MSE starkly increasing towards the right, and slowly towards any other direction, indicating model bias. Most notably, around $(-4.2, -5.2)$ we observe an out-of-bounds area of low reconstruction loss. Due to this type of visualization, we can easily generate an adversarial anomaly by simply decoding the latent space sample: $\boldsymbol{a} = h((-4.2, -5.2))$. This leads to the adversarial anomaly shown in Figure 2c. The adversarial anomaly shares some features with the digits used for training, but does not resemble any of them specifically, making it a clear false negative. Indeed, this sample fulfills our earlier criterion of $\mathcal{L}_R(\boldsymbol{a}_i, \hat{\boldsymbol{a}}_i) \leq \min_i(\mathcal{L}_R(\boldsymbol{x}_i, \hat{\boldsymbol{x}}_i))$, as for this example $\mathcal{L}_R(\boldsymbol{a}_i, \hat{\boldsymbol{a}}_i) = 0.014$, and $\min_i(\mathcal{L}_R(\boldsymbol{x}_i, \hat{\boldsymbol{x}}_i)) = 8.47$.

We also looked at a simpler example, where we train on the digits 0 and 1 to get a clearer separation of the two classes. In our previous experiments with sigmoid activation functions in Section 4.4.2 we observed that at the intersection of the two modalities some unwanted interpolation can occur. In Figure 2b we can observe the same thing, where at the intersection of the two classes we have a very small area in the latent space with a very low reconstruction loss. The normal data close to this area is however not well reconstructed. By generating an artificial sample from the lowest MSE in this latent space, at $(0.535, -0.353)$, we can find an adversarial anomaly $\boldsymbol{a} = h((0.535, -0.353))$ with $\mathcal{L}_R(\boldsymbol{a}_i, \hat{\boldsymbol{a}}_i) = 0.022$, substantially lower than $\min_i(\mathcal{L}_R(\boldsymbol{x}_i, \hat{\boldsymbol{x}}_i)) = 1.61$. This adversarial anomaly

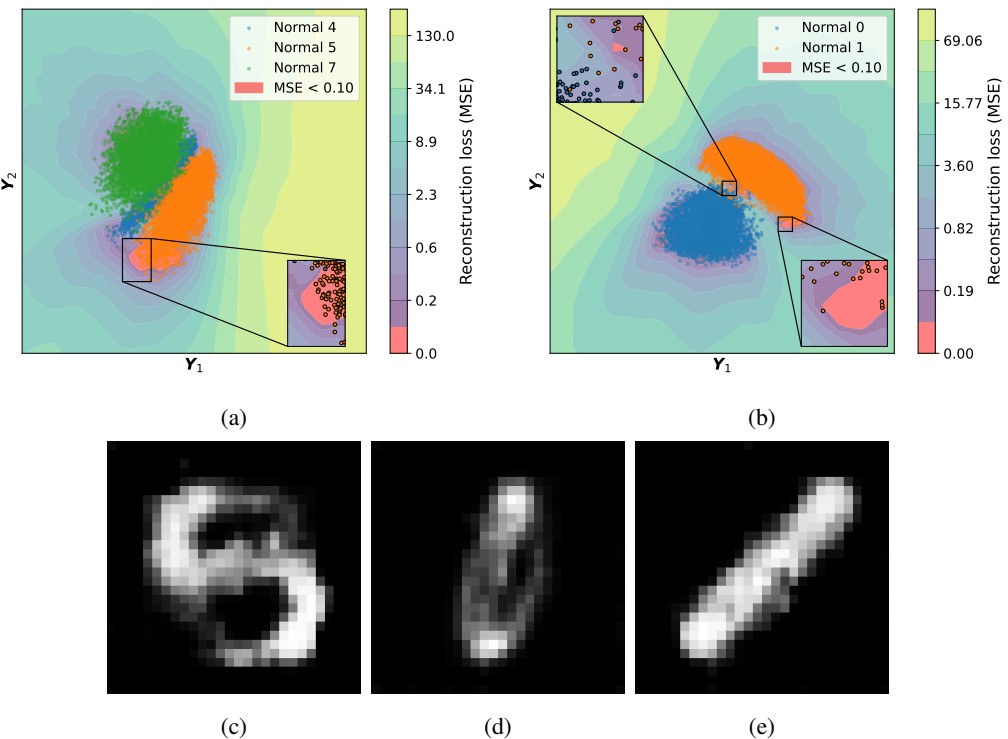

Figure 2: Plots of the contours of the reconstruction loss in the 2D latent space of a convolutional autoencoder when applied on subset of MNIST (a, b), plots of constructed adversarial anomalies (c, d), and a plot of non-problematic out-of-bounds reconstruction (e). Subplots (a, c) show the results for an autoencoder trained on digits 4, 5, and 7, and Subplots (b, d, e) show the results for an autoencoder trained on digits 0, and 1. The visualized samples, i.e. the points in (a, b) are the latent representations of the training data. The shown digits are constructed by sampling from the $\epsilon < 0.1$ zone within the marked area, and correspond to these from left to right. The contourplot is colored red whenever the MSE is below a set threshold $\epsilon < 0.1$ to indicate a near-perfect reconstruction. Note that the color scaling is exponential to better visualize the MSE loss.

is visualized in Figure 2d. We find, unsurprisingly, that the adversarial anomaly here is a mix of the features of the 0 and 1 class.

Similar to our experiments on the digits 4, 5, and 7 autoencoder, we identified an area at the edge of the 1 class where the reconstruction loss is low, but where few normal data points can be found. In contrast to our previous experiment, this area corresponds to a more uncommon diagonally drawn 1, as shown in Figure 2e, which is still within the bounds of what we can consider normal data. From this we can conclude that although out-of-bounds reconstruction can be unwanted, in some cases it aligns with the expectations of an anomaly detector. More generally speaking we observe that some generalization of the autoencoder can align with the expectation of a user. In some cases, generalization can lead to unwanted inter- or extrapolation. This unwanted generalization can cause anomalous data to stay fully undetected. This is similar to the phenomena observed by Nalisnick et al. (2019) for variational autoencoders, who observe some out-of-distribution samples to remain fully undetected.

While conducting the experiments on the MNIST data, we found that the problems shown above do not seem to arise in every case. We observe that depending on the random seed used for initialization and the digits selected as normal data, the out-of-bounds reconstruction may or may not be easily detected. In some cases, the out-of-bounds behavior seems very non-monotonous, meaning that the regions of reconstruction disappear one epoch, and reappear the next. This solidifies our belief that autoencoders may fail, but in many cases outwardly seem to work well. The crux lies in the fact that in a semi-supervised or unsupervised setting, it is not possible to accurately judge whether a

network will fail on future data. This is further complicated by the fact that due to the heterogeneity of anomalies, some may be detected while others go unnoticed.

## 5 CONCLUSION

In this work we provide a theoretical basis of the unwanted reconstruction of anomalies that autoencoders can exhibit when used for anomaly detection. We move beyond existing theories of unwanted reconstruction happening in interpolation and show how unwanted out-of-bounds reconstruction can occur when extrapolating as well, and how this can lead to anomalies staying fully undetected. We show through several experiments that these issues can arise in real-world data and not just in theory. This leads us to some safety concerns, where autoencoders can catastrophically fail to detect obvious anomalies. This unreliability can have major consequences when trust is put into the anomaly detector in safety-critical applications.

In general, we solidify the growing belief that the reconstruction loss is not a reliable proxy for anomaly detection, especially when the network is explicitly trained to lower the reconstruction loss for normal data without constraining the reconstruction capability beyond the bounds of the normal training data such as has been done by Yoon et al. (2021). We find that this issue is most prevalent for (conditionally) linear units such as the ReLU, but similar issues exist for sigmoid networks, albeit to a lesser degree. The reconstruction issue is mostly caused by the fact that a point in the lower-dimensional latent space corresponds to a hyperplane in the original space that the data occupies. Next to interpolation and out-of-bounds reconstruction, we find that anomalies can remain undetected when they occupy the latent space where normal classes border.

Users of autoencoders for anomaly detection should be aware of these issues. Good practice would be to at least check whether a trained non-linear autoencoder exhibits the undesirable out-of-bounds reconstruction. In this paper's illustrative examples, we checked for this by searching for adversarial anomalies. This was relatively easy, as it could be done either visually in the latent space, or through a simple 2D grid search. For more complex datasets, requiring larger latent spaces, a feasible strategy might be to again synthesize samples from the latent space and formulate the search for adversarial anomalies as an optimization in terms of projected gradient descent (Madry et al., 2017).

By describing exactly how autoencoders are unreliable anomaly detectors by describing anomaly reconstruction, we hope to provide a scaffold for future research into fixing and avoiding the identified issues in a targeted manner.

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

# A    APPENDIX

## A.1    LINEAR NETWORKS WITH BIAS TERMS

Linear neural networks with a bias term, similar to those without a bias term, still exhibit out-of-bounds reconstruction that leads to zero reconstruction loss for certain anomalous data points.

Linear autoencoders with bias terms consist of a single linear encoding layer and a single linear decoding layer, each with an added bias term. Like for linear networks without a bias, all multi-layer networks can be reduced to a single layer autoencoder. At the global optimum the bias terms

will recover the process of mean-centering. Note that a simplified version of this proof was presented by Bourlard & Kamp (1988).

**Theorem 4.** *Let $\bar{x} = \frac{1}{m}\mathbf{1}_m X$, so the vector of length $n$ where each element contains the corresponding column-wise mean of $X$. The reconstruction loss $\mathcal{L}_R(b_{enc}, b_{dec}; X, \hat{X})$ for fixed $W_{enc}$, $W_{dec}^T$ is minimized by $b_{enc} = -\bar{x}W_{enc}$, and $b_{dec} = \bar{x}$.*

*Proof.* First let us acknowledge that

$$\bar{x} = \frac{1}{m}\sum_{i=1}^{m} x_i,$$

and thus

$$\sum_{i=1}^{m}(x_i - \bar{x}) = \mathbf{0}.$$

We can then express the average reconstruction loss over the entire dataset as:

$$\mathcal{L}_R(b_{\text{enc}}, b_{\text{dec}}; X, \hat{X}) = \frac{1}{mn}\sum_{i=1}^{m}|x_i - \hat{x}_i|^2$$

$$= \frac{1}{mn}\sum_{i=1}^{m}|x_i - h(g(x_i))|^2$$

$$= \frac{1}{mn}\sum_{i=1}^{m}|x_i - ((x_i W_{\text{enc}} + b_{\text{enc}})W_{\text{dec}}^T + b_{\text{dec}})|^2$$

$$= \frac{1}{mn}\sum_{i=1}^{m}|x_i - x_i W_{\text{enc}}W_{\text{dec}}^T - b_{\text{enc}}W_{\text{dec}}^T - b_{\text{dec}}|^2$$

$$= \frac{1}{mn}\sum_{i=1}^{m}|x_i(1 - W_{\text{enc}}W_{\text{dec}}^T) - b_{\text{enc}}W_{\text{dec}}^T - b_{\text{dec}}|^2$$

$$= \frac{1}{mn}\sum_{i=1}^{m}|(x_i - \bar{x})(1 - W_{\text{enc}}W_{\text{dec}}^T) + (\bar{x} - \bar{x}W_{\text{enc}}W_{\text{dec}}^T) - b_{\text{enc}}W_{\text{dec}}^T - b_{\text{dec}}|^2$$

$$= \frac{1}{mn}\sum_{i=1}^{m}|(x_i - \bar{x})(1 - W_{\text{enc}}W_{\text{dec}}^T)|^2$$

$$+ \frac{1}{mn}\sum_{i=1}^{m}|(\bar{x} - \bar{x}W_{\text{enc}}W_{\text{dec}}^T) - b_{\text{enc}}W_{\text{dec}}^T - b_{\text{dec}}|^2$$

$$+ \frac{1}{mn}\sum_{i=1}^{m}((x_i - \bar{x})(1 - W_{\text{enc}}W_{\text{dec}}^T))((\bar{x} - \bar{x}W_{\text{enc}}W_{\text{dec}}^T) - b_{\text{enc}}W_{\text{dec}}^T - b_{\text{dec}})$$

$$+ \frac{1}{mn}\sum_{i=1}^{m}((\bar{x} - \bar{x}W_{\text{enc}}W_{\text{dec}}) - b_{\text{enc}}W_{\text{dec}}^T - b_{\text{dec}})((x_i - \bar{x})(1 - W_{\text{enc}}W_{\text{dec}}^T))$$

$$= \frac{1}{mn}\sum_{i=1}^{m}|(x_i - \bar{x})(1 - W_{\text{enc}}W_{\text{dec}}^T)|^2$$

$$+ \frac{1}{mn}\sum_{i=1}^{m}|(\bar{x} - \bar{x}W_{\text{enc}}W_{\text{dec}}^T) - b_{\text{enc}}W_{\text{dec}}^T - b_{\text{dec}}|^2.$$

Notice that the left term is constant with respect to $b_{\text{dec}}$ and $b_{\text{enc}}$, and the right term is minimized when $\frac{1}{mn}\sum_{i=1}^{m}|(\bar{x} - \bar{x}W_{\text{enc}}W_{\text{dec}}^T) - b_{\text{enc}}W_{\text{dec}}^T - b_{\text{dec}}|^2 = 0$. If we now substitute $b_{\text{enc}} = -\bar{x}W_{\text{enc}}$,

and $b_{\text{dec}} = \bar{x}$:

$$\frac{1}{mn} \sum_{i=1}^{m} |(\bar{x} - \bar{x} W_{\text{enc}} W_{\text{dec}}^T) - b_{\text{enc}} W_{\text{dec}}^T - b_{\text{dec}}|^2 =$$

$$\frac{1}{mn} \sum_{i=1}^{m} |(\bar{x} - \bar{x} W_{\text{enc}} W_{\text{dec}}^T) + \bar{x} W_{\text{enc}} W_{\text{dec}}^T - \bar{x}|^2 = 0$$

thereby showing that the optimal solution for the biases indeed recovers the process of mean centering.

$\square$

Also note that the other term now mimics the reconstruction loss on the mean-centered data. This means that we find $V_d$ by performing PCA not on $X$, but on $(X - \bar{X})$. Again, we can use the same strategy $a = cV_d^T$ to find adversarial anomalies.

