# OpenReview forum: "Autoencoders for Anomaly Detection are Unreliable"
_ICLR.cc/2025/Conference — ICLR 2025 Conference Withdrawn Submission_

### Official Review · Reviewer_pYYF · 2024-10-28

**Soundness:** 2
**Presentation:** 3
**Contribution:** 2
**Rating:** 6
**Confidence:** 3

**Summary:**

This paper explains that when autoencoders (AEs) are used for anomaly detection, their reliability can be compromised because they may reconstruct data outside of the training dataset. First, this paper proves that PCA and linear AE can reconstruct data not included in the training dataset. Then, it demonstrates that this issue also occurs in nonlinear cases and when using convolutional layers.

**Strengths:**

- This paper theoretically and visually demonstrates that autoencoders can reconstruct data that is not included in the training dataset.

**Weaknesses:**

- I am unclear about the novelty of this paper. Please refer to the Questions section.

**Questions:**

- Could you tell me the contributions of this paper? The paper theoretically proves that PCA and linear autoencoders (AEs) can reconstruct anomalies, but does this qualify as novelty? This issue is widely known, and for example, [1] demonstrates that variational autoencoders (VAEs) cause this problem. Although there may be probabilistic or deterministic differences, I believe that AEs are a special case of VAEs, so this should be mentioned in the paper. From the perspective of probabilistic models like VAEs, for instance, [2] provides a theoretical explanation of this problem. Could you explain the relationship between these previous studies and this paper?

- For example, in the case of VAEs, this issue can be mitigated by using hierarchical models as in [3] or by employing ensembles as in [4]. For autoencoders, this problem can be mitigated by restricting latent variables as in [5] or by normalizing the latent space by considering it as an Energy-Based Model (EBM), as in [6]. In this paper, specific anomalies that can potentially be reconstructed are identified. Based on these insights, could you propose any ideas on how to mitigate this issue?

[1] Nalisnick, Eric, et al. "Do deep generative models know what they don't know?." arXiv preprint arXiv:1810.09136 (2018).

[2] Fang, Zhen, et al. "Is out-of-distribution detection learnable?." Advances in Neural Information Processing Systems 35 (2022): 37199-37213.

[3] Havtorn, Jakob D., et al. "Hierarchical vaes know what they don’t know." International Conference on Machine Learning. PMLR, 2021.

[4] Choi, Hyunsun, Eric Jang, and Alexander A. Alemi. "Waic, but why? generative ensembles for robust anomaly detection." arXiv preprint arXiv:1810.01392 (2018).

[5] Zhou, Yibo. "Rethinking reconstruction autoencoder-based out-of-distribution detection." Proceedings of the IEEE/CVF Conference on Computer Vision and Pattern Recognition. 2022.

[6] Yoon, Sangwoong, Yung-Kyun Noh, and Frank Park. "Autoencoding under normalization constraints." International Conference on Machine Learning. PMLR, 2021.

---

> ### Author Response · Authors · 2024-11-25
>
> Dear reviewer,
>
> Thank you for your valuable review and questions.
> We will further aim to answer your questions to the best of our ability:
> - While this work is not the first to identify that autoencoders can reconstruct anomalies, it is to the best of our knowledge the first that explores the unwanted property of anomaly reconstruction from both a theoretical and practical perspective. To relate this to the literature you supplied: [1] Indeed note the problem for VAEs within the context of OOD detection, but do not go in depth as to why it happens. [2] provides a theoretical explanation of the learnability of OOD by proving mathematically that OOD detection is unlearnable if the ID and OOD groups overlap. While we were unaware of the paper and the proof it contains, we think it is quite interesting. In this context, we can relate their proof to ours: If the ID and OOD groups overlap in the latent space, OOD detection should become unlearnable.
> - In [3] an interesting explanation is offered for how VAEs learn low level features that are highly correlated between ID and OOD datasets. From our proofs and observations it seems that is related to our theory on how anomaly reconstruction can happen. Indeed the low level features that are found can then lead to a mapping on the hyperplane of (near-)perfect reconstruction. In this was, it seems like both our explanations offer complementary insights. [4] seems only tangentially related, as they only tackle the likelihood-based detection scenario, and not the reconstruction-based scenario which we consider. [5] Again considers the reconstruction-based scenario. They however offer the explanation that OOD data has smaller neural activations. We think this is indeed a good observation based on the common examples found in OOD detection literature. In the context of the perfect-reconstruction hyperplane, this is related to the multi-cluster interpolation scenario discussed in our work. In the case of small neural activations, the OOD data can occupy the hyperplane in between or just outside the ID data, whilst having smaller neural activations than the normal data itself. Perhaps the most interesting work which we were previously unaware of is [6]. Indeed they offer an explanation much along the lines of ours: they theorize that anomalies can occupy the low-reconstruction manifold, and therefore fail to be detected. In their appendix they show this, and also relate it to the classical result of Bourlard & Kamp (1988), relating autoencoders to PCA. It should be noted here that the example that is given only concerns the case of between-cluster interpolation, while in our work we provide multiple examples of unwanted extrapolation/reconstruction outside of the convex hull of the normal data.
>
> In this review you have provided several useful insights which we will reflect on in our updated version of the paper. You have furthermore brought several papers to our attention that we were unaware of. We will incorporate these into an updated version of this work.
> In summary, we feel that, through experimentation and proofs, we provide new insights into the phenomenon of anomaly reconstruction and the failure of reconstruction-based autoencoders.

---

> > ### Comment · Reviewer_pYYF · 2024-11-26
> >
> > Thank you for your reply.
> > I now have a better understanding of how this relates to existing research.
> >
> > I think it is worth noting that you have theoretically demonstrated why autoencoders fail in anomaly detection.
> > However, I believe it is crucial to clarify the links to other research and discuss them more thoroughly in the paper.
> >
> > I will maintain my score.

---

> > > ### Author Response · Authors · 2024-11-26
> > >
> > > Dear reviewer,
> > >
> > > Thank you for your timely response.
> > > We agree that it is essential to further solidify our links to related work. This will be reflected in the updated version of the manuscript which will include a greatly extended introduction and related work section.

---

> > > > ### Comment · Reviewer_pYYF · 2024-12-03
> > > >
> > > > I apologize for the delayed response.
> > > > It seems that the revised paper has clarified its relationship with existing research.
> > > > While I believe there is still room for improvement,
> > > > considering its significance,
> > > > I raise my score to 6.

---

### Official Review · Reviewer_LMNo · 2024-11-02

**Soundness:** 2
**Presentation:** 2
**Contribution:** 1
**Rating:** 3
**Confidence:** 5

**Summary:**

This paper discusses that an autoencoder may reconstruct anomalies even though the anomaly resides very far from the training data. Their analysis starts with the linear case, i.e., PCA, and then is extended to neural networks, showing that an autoencoder can give a very small reconstruction error for outliers. However, the paper does not provide a solution to remedy this phenomenon.

**Strengths:**

* The paper discusses the under-appreciated problem of the autoencoder being capable of generating anomalies when applied to anomaly detection.
* The overall exposition of the paper is clear and easy to follow.

**Weaknesses:**

* The paper only reports the problem but not a solution. The contribution of the paper is questionable, as the unexpected reconstruction of anomalies by an autoencoder was mentioned and studied several times in previous works. According to line 427 of the manuscript, this work is not the first to report the reconstruction of anomalies.
* There are missing references that reported and discussed the anomaly reconstruction phenomenon.
    * https://uvadlc-notebooks.readthedocs.io/en/latest/tutorial_notebooks/tutorial9/AE_CIFAR10.html#Out-of-distribution-images
    * Autoencoding under normalization constraints https://arxiv.org/abs/2105.05735 and references therein. Their appendix contains analyses similar to those provided by the manuscript.
    * Outlier reconstruction web demo https://swyoon.github.io/outlier-reconstruction/
* The value of the analyses provided by the paper is not clear. Most sections are dedicated to simply showing the existence of reconstructed anomalies, which is somewhat trivial. The analyses do not lead to deeper insight, which can be used to build better anomaly detection algorithms.

**Questions:**

See weaknesses.

---

> ### Author Response · Authors · 2024-11-25
>
> Dear reviewer,
>
> We want to express our thanks for your review.
> We will reply to each of your listed comments below:
> - "The paper only reports the problem but not a solution." The focus of our paper is indeed on providing further insights into possible failure of autoencoders when applied to anomaly detection. This is very much intended, by not aiming to provide a solution, but rather describing the problem to the fullest possible extent, we provide an unbiased view of what "needs to be fixed". The paper therefore functions as a cautionary tale, but also as a scaffold for future research in potentially fixing the listed problems. This is in line with similar literature published at ICLR and similar conferences in previous years, such as Nalisnick, Eric, et al. "Do Deep Generative Models Know What They Don't Know?." International Conference on Learning Representations, and Reiss, Tal, et al. "Anomaly detection requires better representations." European Conference on Computer Vision. Cham: Springer Nature Switzerland, 2022.
> - Thank you for pointing out these missing references. The second reference and the corresponding web demo demonstrate similar problems to the ones outlined in the paper. We've updated our manuscript to include the observations made in this paper. The first reference however we do not deem fully relevant, as they only discuss the trivially detectable example of pure noise outliers detectable by reconstruction loss, not the perhaps unexpected reconstruction of samples of other datasets or class anomalies.
> - In our analyses we aim to not only show that reconstructed anomalies exist, but also provide a mathematical basis on how this phenomenon can occur. The experiments we've conducted further build on this, by not only showing that there is failure, but also describing the properties of the reconstructed anomalies and the latent space of the autoencoder. We think that these new insights can explicitly contribute to the development of better anomaly detection algorithms. We've updated our manuscripts to better reflect these thoughts.
>
> We will update the manuscript accordingly to reflect the feedback and our response to it.
> We want to thank the reviewer again for the constructive feedback.
>
> Kind regards,

---

> > ### Comment · Reviewer_LMNo · 2024-11-26
> > **Thanks for the response**
> >
> > Dear authors,
> >
> > I appreciate your answers to the review. I acknowledge that this paper points out the significant failure mode of autoencoder-based anomaly detection. However, given that this problem has already been discussed in earlier works and the paper does not take a step forward to build a solution for it, I am afraid the current version of the manuscript is not sufficient to be presented at the conference.

---

### Official Review · Reviewer_KgmR · 2024-11-04

**Soundness:** 1
**Presentation:** 3
**Contribution:** 2
**Rating:** 3
**Confidence:** 4

**Summary:**

This paper demonstrates that using auto-encoder reconstruction loss is an ineffective metric for anomaly detection. It argues that it is possible to generate instances that lie on the PCA hyperplane yet are still anomalous.

This argument is valid and straightforward to illustrate. Intuitively, the PCA hyperplane has an infinite volume, whereas the volume occupied by normal examples is finite. Therefore, there must exist points on the PCA hyperplane that are anomalous. The same reasoning applies to non-linear auto-encoders as well.

The authors then extend their conclusion to state that "Auto-encoders are unreliable for anomaly detection." However, this generalization is incorrect because the unreliability stems from the authors' application of PCA rather than the PCA methodology itself.

The core assumption of PCA is that there is an underlying multivariate Gaussian distribution. Consequently, PCA provides variance estimates along the principal components (the eigenvalues). These variance estimates can be utilized to calculate likelihoods, which are valuable for anomaly detection.

Other autoencoders either output variance directly (such as VAEs) or allow for the estimation of variance and covariance through multiple examples.

The unreliability of reconstruction error for anomaly detection, as discussed by the authors, arises from disregarding the variance information provided by PCA, not from the PCA process itself.

Many anomaly detection studies employ autoencoders and model the latent space with Gaussian assumptions without encountering the issues highlighted in this paper. Examples include:

Tipping, M. E., & Bishop, C. M. (1999). "Probabilistic Principal Component Analysis."
Archambeau, C., et al. (2007). "Anomaly Detection with Robust Deep Autoencoders."
Xie, J., Girshick, R., & Farhadi, A. (2016). "Unsupervised Deep Embedding for Clustering Analysis."

Although I find the mathematical foundations and arguments presented in the paper to be sound. However, the arguments whereby auto-encoders are unreliable, are not sound.

**Strengths:**

+ Mathematical notations and flow are sound.

**Weaknesses:**

- Discussed in the summary.

**Questions:**

.

---

> ### Author Response · Authors · 2024-11-15
> **Additional questions and comments to the reviewer based on their review**
>
> Dear reviewer,
>
> We want to thank you for your review and comments. We've updated our manuscript based on some of your suggestions, but would also kindly request some clarification of some of your comments.
>
> 1. The reviewer mentions that PCA assumes a multivariate Gaussian distribution. To the best of our knowledge, this is not true, but it rather assumes just linear data. See for example https://arxiv.org/abs/1404.1100. Some confusion may arise from the Probabilistic PCA context, where a linear-Gaussian is assumed over the latent variables, not the observed variables. In fact, in a linear-Gaussian framework, in which the marginal and conditional distributions are Gaussian (see also Pattern Recognition and Machine Learning by Bishop, section 12.2). Could the reviewer clarify whether our interpretation is now correct?
> 2. The reviewer states that our reasoning is flawed due to not using the variance information in the PCA model. Does the reviewer mean that when applying for example probabilistic PCA, a likelihood over the latent variables can be calculated? If so, we would like to stress that PCA is only used as a scaffold in our proof for autoencoders. We furthermore understand that there may be better ways than just using the reconstruction loss, but the reconstruction loss is still applied widely in practice.
> 3. The reviewer states that "Other autoencoders either output variance directly (such as VAEs) or allow for the estimation of variance and covariance through multiple examples.". Indeed, some methods such as VAEs or PCA can explicitly model the covariance of the data. If I understand the reviewer correctly, they refer to for example the use of Hotelling's T2 in outlier detection in the PCA space, to the explicit constraints put on the the latent space in VAEs. Is that correct?
> 4. The reviewer then states: "The unreliability of reconstruction error for anomaly detection, as discussed by the authors, arises from disregarding the variance information provided by PCA, not from the PCA process itself." We respectfully disagree here. While the variance information can help in some cases, it certainly does not guarantee outlier detection in the way we state is our objective. For example in "Do Deep Generative Models Know What They Don't Know?" by Nalisnick, it is made clear that VAEs, even while allowing for anomaly detection using the likelihood, still suffer from the inability to detect obvious anomalies. This is further solidified in for example "Autoencoding under normalization constraints by Yoon et al (ICML 2021) "
> 5. The reviewer further mentions "Anomaly Detection with Robust Deep Autoencoders" by Archambeau et al (2007). I was unable to find the referenced paper, but I believe the reviewer might refer to "Anomaly Detection with Robust Deep Autoencoders by Zhou and Paffenroth (KDD 2017)". However, this paper to my understanding does not use the latent space for anomaly detection, but rather aims to minimize the influence of anomalies in the training of the autoencoder, much like robust PCA, by splitting the data into normal and anomalous data.  Interestingly, the results of this paper showcase the failure of normal autoencoders, further solidifying the need for analysis of this phenomenon. Could the reviewer comment on whether the suggested paper is indeed the one they meant to suggest?
> 7. The reviewer mentions that "Many anomaly detection studies employ autoencoders and model the latent space with Gaussian assumptions without encountering the issues highlighted in this paper. Examples include: ... " However, of these studies, the first and third do not perform any anomaly detection at all. Furthermore, we specifically study the setting where an autoencoder is used in conjunction with just the reconstruction loss, a practice which is common throughout industry and academia, as is outlined in the introduction of our manuscript. Could the reviewer elaborate in more detail on the relevance of these studies on our manuscript?
>
> We've also further clarified the goal of our work, in which we agree with the reviewer, namely: using an autoencoder with the reconstruction loss is insufficient to guarantee reliable anomaly detection. We prove and show how autoencoders can fail, and thereby aim to fill the theoretical gap in our understanding of autoencoder methods when used for anomaly detection.

---

### Official Review · Reviewer_G4ux · 2024-11-04

**Soundness:** 3
**Presentation:** 3
**Contribution:** 3
**Rating:** 6
**Confidence:** 3

**Summary:**

This paper examines the limitations of Autoencoders for anomaly detection from a theoretical perspective. It explores approaches including PCA, linear Autoencoders, non-linear Autoencoders, and Convolutional Autoencoders, using synthetic tabular data and the MNIST dataset. The study demonstrates that under certain conditions, anomalies can yield zero or low reconstruction errors in autoencoders, leading to the conclusion that reconstruction loss may not reliably indicate anomalies.

**Strengths:**

- The paper presents a novel and intriguing perspective by questioning the use of autoencoders for anomaly detection theoretically.
- Analysis is conducted through both theoretical exploration and practical examples using synthetic data and MNIST.
- Figure 1 results are well-illustrated and offer interesting findings.

**Weaknesses:**

- Although this work highlights the limitations of autoencoders for anomaly detection, it may not fully address practical cases where anomalies come from out-of-class data or shifts in distribution, which differ from the conditions presented. For instance, in Figure 2, some examples show failure cases in convolutional autoencoders, but in many regions, reconstruction errors appropriately increase as the distribution shift grows. Specifically, in Figure 2(a), the presence of low reconstruction errors might be reasonable, as these cases are close to the normal training data and could be considered examples of generalization.

**Questions:**

* Could the authors provide more discussion about how the conclusion can be generalized in the out-of-class anomalies or other potential realistic distribution shifts or anomaly cases?

---

> ### Author Response · Authors · 2024-11-25
>
> We want to thank the reviewer for their comments on this manuscript. We'd like to address the listed weaknesses in order to further improve this paper.
>
> **Weaknesses**
> The reviewer notes correctly that in Figure 2(a) the reconstruction errors grows appropriately in many regions. This desirable behaviour is sadly not present in all regions, as illustrated in the same figure in the highlighted section.
> As the reviewer notes, the region of low reconstruction loss with no normal training data in this highlighted section can be seen as an example of generalization. While in many applications we want models to generalize, it can be counterproductive for anomaly detection. When an anomaly detector generalizes too well, it can assign a low recontruction loss to clear anomalous samples, as is the case for the synthesized samples in Figure 2(c-e). It should be noted that for all of these synthesized samples, the reconstruction loss is lower than for all normal data, which even violates the assumption of wanted generalization. When exactly generalization becomes unwanted is of course highly domain dependent, but the crux lies in the fact that this behaviour can happen, and can lead to undetected samples.
> We've updated our manuscript to more clearly reflect the insights of the reviewer, as well as our explanation of the observed phenomena in this light.
>
> **Questions**
> As for the last question posed by the author: "Could the authors provide more discussion about how the conclusion can be generalized in the out-of-class anomalies or other potential realistic distribution shifts or anomaly cases?":
> - The question asked by the reviewer is a fair one. We'd summarize it as follows:
> 	- due to unwanted generalization of an autoencoders, various anomalies, whether they be out-of-class or out-of-distribution, can stay undetected due to having lower reconstruction errors than normal data.
> 	- When the distribution of data shifts realistically, either of two scenarios can happen: 1) the shifts happens into an area of low reconstruction error, indicating that the autoencoders exhibits wanted generalization, or 2) the shift happens into an area of higher than normal reconstruction error, leading to potential false positives by the detector. In this situation, there is still no guarantee that potential anomalies won't occupy the areas of low reconstruction loss, which is unwanted generalization.
> We will update the manuscript to include this introspection.

---

> > ### Comment · Reviewer_G4ux · 2024-12-02
> >
> > Thank you for the authors' rebuttal. I have reviewed the authors’ responses and carefully considered the discussion between the authors and other reviewers. I believe that further discussion and comparison with prior works could enhance the quality of the paper. I will maintain my current score.

---

### Author Response · Authors · 2024-11-27
**Revised manuscript**

Dear reviewers,

We want to sincerely thank you for your reviews and insights. Based on your valuable feedback, we've revised our manuscript. We hope to have now addressed all of your concerns sufficiently. Detailed responses to each of your questions and listed weaknesses can be found in each individual response thread.
More generally speaking, the greatest changes in our manuscript are the following:
- We've greatly and carefully extended our literature review based on the suggestion provided by you, the reviewers. During this extended study, we've also found several other relevant papers not directly mentioned. All of the insights following from this can be found in the updated paper.
- We've updated the framing of our paper to better discern between the current state of the field, and our individual contribution to it.

We want to thank you again for your input.

Kind regards,

---

### Note · Authors · 2025-01-23

I have read and agree with the venue's withdrawal policy on behalf of myself and my co-authors.